# DiffTester: Accelerating Unit Test Generation for Diffusion LLMs via Repetitive Pattern

## Abstract

Software development relies heavily on extensive unit testing, which makes the efficiency of automated Unit Test Generation (UTG) particularly important. However, most existing LLMs generate test cases one token at a time in each forward pass, which leads to inefficient UTG. Recently, diffusion LLMs (dLLMs) have emerged, offering promising parallel generation capabilities and showing strong potential for efficient UTG. Despite this advantage, their application to UTG is still constrained by a clear trade-off between efficiency and test quality, since increasing the number of tokens generated in each step often causes a sharp decline in the quality of test cases. To overcome this limitation, we present DiffTester, an acceleration framework specifically tailored for dLLMs in UTG. The key idea of DiffTester is that unit tests targeting the same focal method often share repetitive structural patterns. By dynamically identifying these common patterns through abstract syntax tree analysis during generation, DiffTester adaptively increases the number of tokens produced at each step without compromising the quality of the output. To enable comprehensive evaluation, we extend the original TestEval benchmark, which was limited to Python, by introducing additional programming languages including Java and C++. Extensive experiments on three benchmarks with two representative models show that DiffTester delivers significant acceleration while preserving test coverage. Moreover, DiffTester generalizes well across different dLLMs and programming languages, providing a practical and scalable solution for efficient UTG in software development. Code and data are publicly available at https://anonymous.4open.science/r/DLM4UTG.

## 1 Introduction

Unit testing plays a vital role in software development, ensuring that a functionally discrete program unit (*e.g.,* a method) behaves correctly and meets the intended design expectations (Olan, 2003; Gren & Antinyan, 2017). However, manually writing high-quality Unit Tests (UTs) is often extremely time-consuming and labor-intensive (Runeson, 2006), especially in large-scale software development scenarios (Shang et al., 2025). To reduce the manual burden, recent research has increasingly sought many automated approaches to generate UTs (Chipounov et al., 2011; Tufano et al., 2020; Fraser & Arcuri, 2011). Among these, advanced Large Language Models (LLMs) have rapidly become the mainstream solution for automated Unit Test Generation (UTG), due to their strong code understanding and code generation abilities (Wang et al., 2024; Bhatia et al., 2024).

In large-scale software development projects, it is often necessary to generate a substantial number of UTs, sometimes reaching hundreds or thousands (Robinson et al., 2011; Li et al., 2006). However, most existing LLMs (Hui et al., 2024; Achiam et al., 2023) generate UTs one token at a time during each forward pass, which substantially increases both time consumption and computational cost, leading to inefficient UTG (Yang et al., 2024). Fortunately, emerging Diffusion Large Language Models (dLLMs) (Nie et al., 2025b; Ye et al., 2025) exhibit strong potential in UTG and other code-related tasks (Li et al., 2025a; Khanna et al., 2025), which adopt a unique generation paradigm that naturally enables *multi-token prediction* and *flexible generation order*. Specifically, at each inference step, dLLMs predict a candidate token for every `[MASK]` position and then remask those with relatively low confidence according to a remasking strategy.

Although dLLMs are theoretically capable of generating multiple tokens in each forward pass, existing studies (Zeng et al., 2025; Barr et al., 2014) typically set the number of tokens generated per

step to a very small value (*e.g.,* one or two). As a result, their inference speed is often comparable to that of non-diffusion LLMs, which fail to fully exploit the efficiency advantages of dLLMs in theory (Li et al., 2025a). Our preliminary experiments further reveal that when the number of tokens generated per step is increased, the quality of the generated UTs degrades sharply, and in many cases, even the syntactic correctness of the test cases cannot be guaranteed (for more details, see Appendix C.2.3). This limitation greatly hinders the efficiency of dLLMs in UTG, highlighting the need for an acceleration framework specifically tailored to dLLMs for UTG.

To accelerate dLLMs in UTG, we conduct an in-depth analysis of the characteristics of this task. We find that for the same focal method, the generated UTs often share repetitive patterns, frequently exhibiting substantial structural and syntactic repetition. For instance, two generated unit tests often share similar patterns and differ only in specific details. From the perspective of their Abstract Syntax Trees (ASTs) (Peacock et al., 2021), these differences typically appear only in certain leaf nodes or low-level non-leaf nodes. Based on this insight, we propose DIFFTESTER to accelerate unit test generation for diffusion LLMs via repetitive pattern. Specifically, we first prompt dLLMs within a single batch to produce multiple UTs for a given focal method. At selected steps of the generation process, we parse the generated code into multiple ASTs and extract the common nodes across them, which we regard as indicative of patterns inherent to the test cases for the focal method. The tokens corresponding to these common nodes are then generated in a single step. In this way, DIFFTESTER dynamically and appropriately increases the number of tokens generated at each step, while preserving the quality of the resulting test cases.

We perform extensive experiments to validate the effectiveness of our proposed DIFFTESTER on the `TestEval` benchmark (Wang et al., 2025) (whose focal methods are all implemented in Python) with two representative dLLMs, including `Dream` (Ye et al., 2025) and `DiffuCoder` (Gong et al., 2025). To enable comprehensive evaluation and to avoid potential data contamination (Deng et al., 2023), we further extend the `TestEval` benchmark to multiple programming languages, resulting in `TestEval-C++` and `TestEval-Java`. Experimental results demonstrate that, under the same runtime or computational budget, applying DIFFTESTER substantially improves coverage. From another perspective, to achieve the same level of coverage, DIFFTESTER can often reduce both time and computational cost by more than half. We believe that DIFFTESTER effectively accelerates dLLMs in UTG and can contribute to software quality assurance.

## 2 BACKGROUND AND MOTIVATION

### 2.1 INFERENCE PROCESS OF dLLMs

In this section, we introduce how mainstream dLLMs, such as `Dream` (Ye et al., 2025) and `LLaDa` (Nie et al., 2025b), perform inference. These models progressively refine a sequence consisting of $L$ special `[MASK]` tokens into the final generated output (*e.g.,* some test cases). Formally, let $\mathcal{Y}^0 = (y_i^0)_{i=1}^L$, where $y_i^0 = $ `[MASK]`, denote the initial fully masked sequence. Given a prompt $p$, the dLLM performs $T$ steps, each consisting of a forward pass, and eventually produces the output $\mathcal{Y}^T = (y_i^T)_{i=1}^L$, which contains no remaining `[MASK]` tokens.

At each step $t$, the dLLM performs one forward pass to predict the probability distribution for every position that is filled with a `[MASK]` token in parallel. For each such position $i$, the model then samples a non-`[MASK]` token $\hat{y}_i^t$ from its corresponding probability distribution, a process analogous to the sampling procedure used in autoregressive LLMs. After this step, the model proceeds with the remasking operation (Nie et al., 2025b; Ye et al., 2025), where a confidence score is assigned to each predicted token (*e.g.,* based on entropy, maximum probability of the distribution, or by randomly specifying a value). Tokens with the highest confidence are retained, for instance, by selecting the top-$k$ predictions or by keeping those whose confidence exceeds a predefined threshold (Wu et al., 2025a; Wei et al., 2025), while other tokens are reverted back to `[MASK]` for further refinement in subsequent steps. Many prior studies (Li et al., 2025a; Wu et al., 2025a) have shown that increasing the number of tokens generated at each step can significantly degrade the quality of the outputs.

### 2.2 DIVE INTO UNIT TESTS GENERATION

Given a specific focal method, unit test generation typically needs to produce multiple test cases, each consisting of a corresponding test prefix and a test assertion (Lemieux et al., 2023; Zamprogno

**Structural Consistency :**

```java
public void test_shortestPath() {
    Solution solution = new Solution();
    int[][] grid = {{0,0,1},{1,1,0},
                    {0,0,0},{0,1,0}};
    assertEquals(solution.shortestPath
                         (grid, 1), 6);
}

public void test_shortestPath() {
    Solution solution = new Solution();
    int[][] grid = {{0,0},{1,1},{0,0,0}};
    assertEquals(solution.shortestPath
                         (grid, 1), 4);
}
```

**Inline Literals and External Variable References :**

```java
assertEquals(solution.minCost(30, new int[][] {{0, 1, 1},
{1, 2, 3}, {2, 3, 10}}, new int[] {0, 1, 2, 3}), 6);

assertEquals(solution.minCost(maxTime, edges,
passingFees), expectedOutput);
```

**Contains Syntax Errors :**

```cpp
vector<vector<int>> testRoads = {{0, 1, 1}, {1, 2, 2},
{1, 3, 3}};

vector<vector<int>> testRoads = {{0, 1, 3}, {0, 2, 1},},
2 4, },}};
```

Figure 1: Repetitive structural and syntactic patterns frequently emerge in unit test cases generated at an intermediate step of dLLM inference before remasking.

et al., 2022). The test prefix is primarily used to construct the testing data and environment, whereas the test assertion serves to verify the correctness of the focal method (Yuan et al., 2024).

We observe that, for the same focal method, the generated test cases often share repetitive patterns, frequently exhibiting substantial structural and syntactic similarities. As illustrated in the left part of Figure 1, two test cases for one focal method may have almost identical syntactic structures, differing only in the values of variables and constants which determine diversity. We attribute these shared patterns to the strict constraints imposed on both inputs and outputs of the focal method. For example, when the input is a bipartite graph represented by a specific data structure and the output is a list of a given length, the test prefix is structurally constrained by the input, while the test assertion is structurally constrained by the output.

We further find that such structural repetition is widely distributed. Even when test cases employ different styles of construction, such as inline literals or external variable references, as shown in the upper-right part of Figure 1, obvious structural repetition can still be observed. More strikingly, as shown in the lower-right part of Figure 1, even test cases that contain syntax errors—which are common in the intermediate test cases generated at a dLLM inference step before remasking—still exhibit obvious shared patterns across different cases.

We argue that these shared patterns should ideally be generated together when a dLLM is used to produce multiple test cases for the same focal method. However, existing remasking strategies, whether based on selecting the top-$k$ tokens or retaining those above a confidence threshold, fail to exploit the structural repetition intrinsic to unit test generation, ultimately resulting in suboptimal generation efficiency.

✒ **TAKEAWAY:** For the same focal method, the generated test cases often shared repetitive patterns, which we argue can be leveraged to accelerate dLLMs for UTG.

## 3 DIFFTESTER

We propose DIFFTESTER, an approach designed to accelerate dLLMs in unit test generation. The key idea is to mine shared code patterns observed in multiple unit test cases generated at intermediate inference steps and then leverage them to guide token generation. We first provide a brief overview of the overall acceleration procedure (Section 3.1). Then we provide an in-depth explanation of how repetitive patterns are exploited, which forms the core of our approach (Section 3.2). Finally, we introduce several additional techniques to further accelerate inference and ensure the quality of the generated test cases (Section 3.3). An overview of DIFFTESTER is presented in Figure 2.

### 3.1 OVERVIEW OF DIFFTESTER

We detail the overview of DIFFTESTER in this section. For a given focal method, we set the batch size to $n$ (*e.g.*, $n = 3, 5, 7$) and prompt the dLLM to generate one unit test case in each instance, resulting in $n$ unit test cases simultaneously.

At one step $t$, for the $k$-th instance in the batch, the model first predicts tokens for all `[MASK]` positions, producing an intermediate response $\mathcal{Y}^{t,k} = (\hat{y}_i^{t,k})_{i=1}^{L}$. We then follow the standard

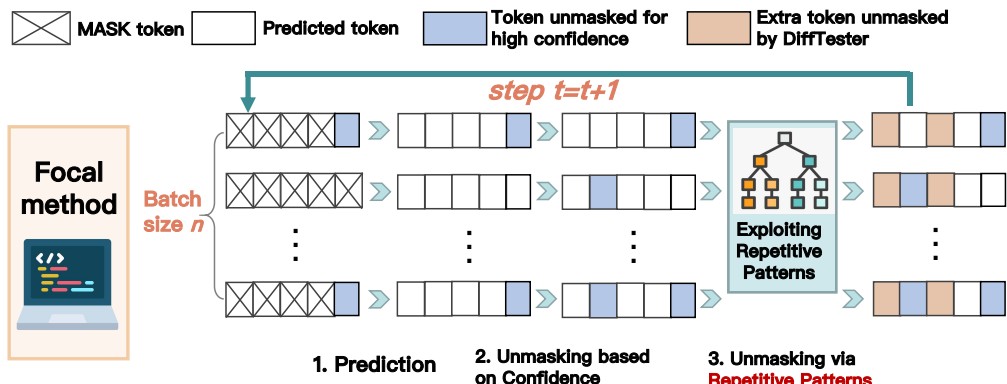

Figure 2: Overview of our proposed DIFFTESTER.

remasking strategy defined in existing dLLMs (*e.g.,* selecting the top-$k$ tokens according to their confidence scores) to determine which tokens will be retained (Ye et al., 2025; Xie et al., 2025; Gong et al., 2025). Subsequently and most importantly, we extract shared patterns across the generated unit test cases and leverage them to further retain additional tokens, with the detailed procedure presented in Section 3.2. After completing the procedure described above, we retain all selected tokens, while the remaining tokens at [MASK] positions are reverted back to [MASK], and the process proceeds to the next step.

## 3.2 EXPLOITING REPETITIVE PATTERNS TO ACCELERATE UTG

In this section, we detail how to exploit repetitive patterns across multiple test cases to accelerate unit test generation for dLLMs.

We begin by analyzing the key challenges involved in leveraging such repetitive patterns, which lie in two main aspects. ❶ The first challenge is **how to effectively extract shared patterns across multiple test cases**. As illustrated in Figure 1, repetitive structural and syntactic patterns can be easily observed among different test cases for the same focal method. However, transforming these intuitive and abstract notions of repetitive patterns into a concrete representation-that is, determining which tokens are structurally repeated—remains a non-trivial problem. ❷ The second challenge is **how to ensure sufficient diversity in the generated test cases**. Achieving high test coverage relies heavily on diversity (Peacock et al., 2021; Yang, 2023), yet the process of extracting shared patterns may risk reducing it. For instance, the model might generate identical input data across multiple test cases. Designing a method that can exploit shared patterns while preserving the diversity of test data is therefore another important challenge.

> ⚠ CHALLENGES: ❶ How to effectively **extract repetitive patterns** across multiple test cases? ❷ How to **ensure sufficient diversity** in the generated test cases?

We next provide a detailed description of how our approach addresses these challenges and ultimately enables the effective utilization of repetitive patterns for acceleration.

**Extract Repetitive Patterns.** We noticed Abstract Syntax Trees (ASTs) provide an objective representation of the syntactic structure of code (Sun et al., 2023; Suttichaya et al., 2022), and each node in an AST can be accurately mapped to a specific set of tokens. Based on this property, our approach compares the ASTs of different test cases' code and attempts to merge them as much as possible, which means identifying the nodes that are shared across different ASTs. When two or more ASTs can be merged into a non-empty tree, this indicates the existence of a repetitive pattern. We then locate the tokens in the intermediate test cases that correspond to the merged AST.

In practice, we find that unit test cases generated at early steps often contain many syntax errors. Such errors propagate through the parsing process, meaning that the same error can have a much greater negative impact when it occurs in a high-level AST (*e.g.,* with the root node being a program) than when it appears in a low-level AST (*e.g.,* with the root node being a statement). As a result, parsing an entire unit test case frequently produces ASTs that fail to accurately reflect the intended code structure and therefore offer little guidance. To address this issue, we construct ASTs at the granularity of individual code lines rather than entire test cases, as illustrated in Figure 3.

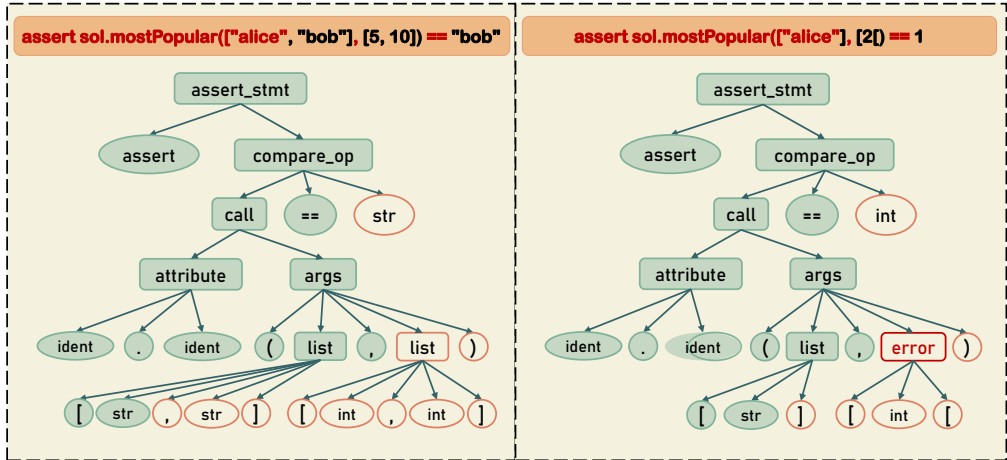

Figure 3: Extract shared nodes between two ASTs and locate their corresponding tokens in the generated code. Square boxes represent non-leaf nodes, while ellipses indicate leaf nodes. The **colored tokens** in the code at the top of the figure highlight the tokens that can be additionally retained according to the merged AST.

**Ensure Sufficient Diversity.** The diversity of unit test cases largely depends on the variability of the input data constructed for the focal method, which in turn is primarily determined by literal values such as integers or floats (Yang, 2023). To preserve this variability, we exclude the AST nodes corresponding to such literal values from the merging process. This design ensures that even if two test cases construct similar data at an intermediate step, the corresponding tokens are not retained in a single step but are instead remasked for subsequent refinement, allowing higher diversity.

After identifying the tokens that belong to the merged AST, we retain them in a single step, while the remaining tokens are remasked for refinement in the subsequent step. The details of the entire process are provided in Algorithm 1 and Algorithm 2 in the Appendix.

### 3.3 ADDITIONAL TECHNIQUES OF DIFFTESTER

We additionally introduce two techniques to improve the quality of the generated test cases and to further accelerate the generation process.

❶ Our preliminary experiments show that directly decoding all tokens belonging to the merged AST can slightly reduce the syntactic correctness of the generated unit test cases. We hypothesize that this is mainly because tokens with very low confidence may occasionally be retained, while in practice, we observe that such cases are extremely rare. To address this issue, we retain only those tokens whose confidence exceeds a predefined threshold $\tau$ when guided by the merged AST, thereby ensuring that the retained tokens are more reliable for generation.

❷ Although parsing code to construct ASTs is computationally inexpensive, invoking the process at every step still introduces noticeable overhead. Fortunately, we observe that the ASTs of code generated in consecutive steps change very little. Based on this observation, we choose not to apply DIFFTESTER at every step, but instead to use it intermittently after several steps, which leads to faster acceleration.

## 4 EXPERIMENT

In this section, we systematically evaluate the performance of DIFFTESTER in accelerating unit test generation. Additional experimental results including ablation studies are provided in Appendix C.2.

### 4.1 EXPERIMENT SETUP

**Models.** We evaluate our approach using two representative dLLMs, namely DIFFUCODER-7B-CPGRPO (Gong et al., 2025) and DREAM-V0-INSTRUCT-7B (Ye et al., 2025). For brevity, we refer to them as `DiffuCoder` and `Dream` in the remainder of this paper.

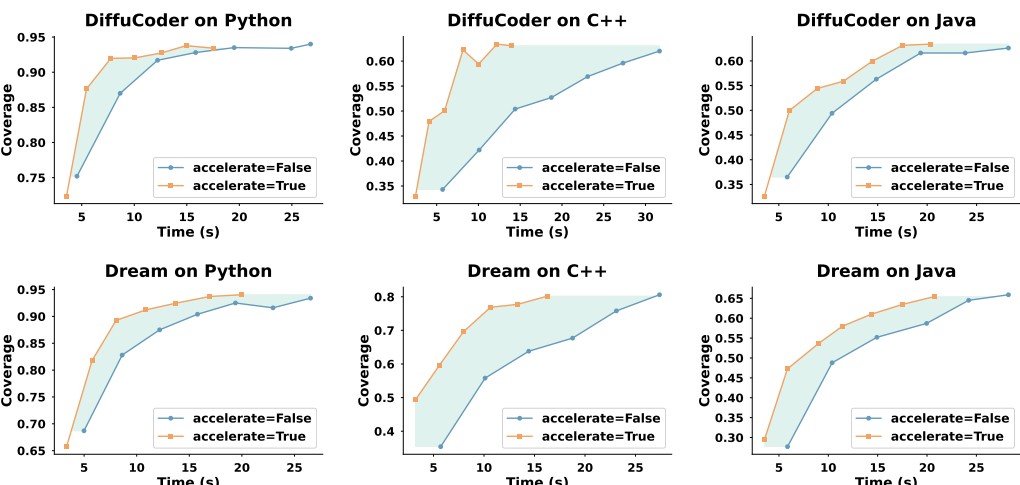

Figure 4: Comparison of line coverage with and without DIFFTESTER at equal decoding time.

**Benchmarks.** To evaluate the performance of DIFFTESTER, we conduct experiments on the `TestEval` benchmark (Wang et al., 2025). `TestEval` is specifically designed to assess the capability of models in unit test generation, and it comprises 210 Python programs collected from LeetCode. We denote this benchmark as `TestEval-Python`. To further validate the generalization ability of DIFFTESTER across different programming languages and to avoid potential data contamination, we extend the `TestEval-Python` benchmark by incorporating two additional programming languages: C++ and Java. Following the construction methodology of `TestEval-Python`, we systematically gather corresponding C++ and Java implementations of the same set of 210 programs. We denote these benchmarks as `TestEval-C++` and `TestEval-Java`, respectively.

**Evaluation Metrics.** We adopt four widely used metrics, three for measuring efficiency and one for assessing test coverage. ❶ *Computational Cost (tflops)*: the average computation per batch. ❷ *Decoding Time (seconds, s)*: the average inference time per batch. ❸ *Throughput (tokens/s, tps)*: the average number of tokens generated per second per batch, excluding special tokens such as `[PAD]` and `[EOS]`. ❹ *Line Coverage*: the average ratio of the number of code lines covered by the generated test cases to the total number of code lines.

**Baseline.** Since there is currently no acceleration approach specifically tailored to dLLMs for UTG, we take dLLMs without applying DIFFTESTER as the baseline.

**Implementation Details.** We set the predefined generation length $L$ to 128 and the confidence threshold $\tau$ to 0.02. In addition, we apply DIFFTESTER once every two steps. Further implementation details are provided in Appendix C.1.3.

### 4.2 MAIN RESULTS

We conduct evaluations on the three benchmarks and two representative models described in Section 4.1. We use different batch sizes $n$, where $n$ ranges from 1 to 7, corresponding to the number of test cases generated per focal method. Figure 4 presents the average test coverage achieved under varying time budgets, while Table 1 reports the efficiency comparison between DIFFTESTER and the baseline when generating the same number of test cases.

**Under the same time budget, DIFFTESTER achieves higher test coverage.** In pipeline of most software development, the time available for unit testing is often limited, which makes maximizing test coverage within a fixed time budget highly desirable. As shown in Figure 4, when the same amount of time is spent generating test cases, DIFFTESTER consistently achieves substantially higher line coverage than the baseline. This demonstrates that the proposed DIFFTESTER is highly competitive and enables dLLMs to achieve higher test coverage more quickly in UTG.

**DIFFTESTER does not compromise the maximum achievable test coverage.** The highest test coverage attainable without time constraints is another important optimization target, particularly for scenarios such as vehicle control systems (Conrad & Fey, 2017; Zhang et al., 2024). We find that DIFFTESTER has little to no negative impact on the maximum test coverage and in some cases

Table 1: Efficiency comparison with and without DIFFTESTER across different batch sizes $n$ on `TestEval-Python`, `TestEval-C++`, and `TestEval-Java`, with `Dream` and `DiffuCoder`.

| Method | $n = 3$ | | | $n = 5$ | | | $n = 7$ | | |
|---|---|---|---|---|---|---|---|---|---|
| | Computational Cost (tflops) | Decoding Time (s) | Throughput (tps) | Computational Cost (tflops) | Decoding Time (s) | Throughput (tps) | Computational Cost (tflops) | Decoding Time (s) | Throughput (tps) |
| *TestEval-Python* | | | | | | | | | |
| DiffuCoder | 1015.59 | 12.22 | 16.97 | 1692.65 | 19.51 | 17.69 | 2369.72 | 26.78 | 18.09 |
| + DIFFTESTER | 580.36 | 7.77 | 26.86 | 997.16 | 12.59 | 27.45 | 1432.00 | 17.57 | 27.42 |
| speedup | ×1.75 | ×1.57 | ×1.58 | ×1.70 | ×1.55 | ×1.55 | ×1.65 | ×1.52 | ×1.52 |
| Dream | 1015.59 | 12.18 | 15.61 | 1692.66 | 19.39 | 15.96 | 2369.72 | 26.53 | 16.68 |
| + DIFFTESTER | 605.05 | 8.04 | 23.59 | 1093.38 | 13.68 | 23.09 | 1644.05 | 19.95 | 22.45 |
| speedup | ×1.68 | ×1.51 | ×1.51 | ×1.55 | ×1.42 | ×1.45 | ×1.44 | ×1.33 | ×1.35 |
| *TestEval-C++* | | | | | | | | | |
| DiffuCoder | 1216.98 | 14.40 | 9.73 | 2098.93 | 23.08 | 9.62 | 2924.4 | 31.68 | 9.58 |
| + DIFFTESTER | 429.82 | 5.95 | 23.81 | 777.32 | 10.02 | 23.60 | 1121.37 | 14.00 | 23.20 |
| speedup | ×2.83 | ×2.42 | ×2.45 | ×2.70 | ×2.30 | ×2.45 | ×2.61 | ×2.26 | ×2.42 |
| Dream | 1216.98 | 14.43 | 13.11 | 2028.30 | 23.11 | 13.45 | 2839.62 | 31.60 | 13.88 |
| + DIFFTESTER | 588.17 | 7.96 | 25.93 | 1041.20 | 13.33 | 25.45 | 1614.85 | 19.97 | 24.82 |
| speedup | ×2.07 | ×1.81 | ×1.98 | ×1.95 | ×1.73 | ×1.89 | ×1.76 | ×1.58 | ×1.79 |
| *TestEval-Java* | | | | | | | | | |
| DiffuCoder | 1259.36 | 14.86 | 16.10 | 2098.93 | 23.85 | 16.69 | 2924.41 | 32.59 | 16.71 |
| + DIFFTESTER | 667.65 | 8.88 | 29.17 | 1143.00 | 14.46 | 28.58 | 1649.17 | 20.33 | 28.20 |
| speedup | ×1.89 | ×1.67 | ×1.81 | ×1.84 | ×1.65 | ×1.71 | ×1.77 | ×1.60 | ×1.69 |
| Dream | 1259.36 | 14.92 | 15.75 | 2098.93 | 24.22 | 15.79 | 2938.51 | 32.64 | 13.58 |
| + DIFFTESTER | 673.40 | 8.97 | 27.76 | 1130.74 | 14.34 | 28.12 | 1678.98 | 20.72 | 27.35 |
| speedup | ×1.87 | ×1.66 | ×1.76 | ×1.86 | ×1.69 | ×1.78 | ×1.75 | ×1.58 | ×2.01 |

even improves it. For example, as shown in Figure 4, on the `TestEval-Python` benchmark with the `Dream` model, applying DIFFTESTER leads to a slight increase in max test coverage.

**DIFFTESTER can effectively improve the efficiency of dLLMs for UTG.** Across nearly all settings, dLLMs achieve more than a 1.5× improvement in efficiency when equipped with DIFFTESTER. For example, with a batch size $n = 3$ on `TestEval-C++` using `DiffuCoder`, DIFFTESTER reduces the computational cost and decoding time from 1217 TFLOPs and 14.4s to 430 TFLOPs and 6.0s, respectively, while increasing throughput from 9.7 TPS to 23.8 TPS. These results demonstrate that DIFFTESTER can substantially accelerate dLLMs in unit test generation.

**DIFFTESTER generalizes well across different models and programming languages.** Across various models and programming languages, DIFFTESTER exhibits consistent acceleration trends and similar effects on coverage. Notably, the acceleration effect of DIFFTESTER is more pronounced on `TestEval-C++` compared to the other two benchmarks. For example, with a batch size of 5 using `DiffuCoder`, throughput improves by 1.55× on `TestEval-Python` and 1.71× on `TestEval-Java`, whereas on `TestEval-C++` it increases by 2.45×. We attribute this to the fact that commonly used syntactic structures in C++ exhibit greater structural repetition than those in Python and Java, which makes it easier to exploit repetitive patterns for acceleration.

## 5 DISCUSSION

### 5.1 COMPARISON WITH dLLM ACCELERATION FOR GENERAL TASK

We observe that several contemporary works have also explored training-free approaches to accelerate dLLMs (Wei et al., 2025; Ben-Hamu et al., 2025; Israel et al., 2025). However, these approaches typically focus on general-purpose tasks, whereas DIFFTESTER is designed specifically for unit test generation (UTG), a domain where efficiency is particularly critical. To highlight the difference, we compare DIFFTESTER with a representative approach, EB-SAMPLER (Ben-Hamu et al., 2025). EB-SAMPLER accelerates dLLM inference through an Entropy-Bounded unmasking procedure, which dynamically unmasks multiple tokens under a predefined approximate error tolerance.

We conduct experiments on the `TestEval-Python`, testing batch sizes of $n = 3, 5, 7$, and report both average decoding time and line coverage in Table 2. ❶ Compared with EB-SAMPLER, DIFFTESTER achieves more effective acceleration in generating unit tests. For example, when $n = 7$, DIFFTESTER reduces the average decoding time to 17.6%, whereas EB-SAMPLER reduces it only to 19.8%. ❷ In addition, the quality of the test cases generated by DIFFTESTER is substantially higher. With

Table 2: Decoding time and line coverage on `TestEval-Python` with batch sizes $n = 3, 5, 7$, comparing the baseline `DiffuCoder`, EB-SAMPLER, and DIFFTESTER.

| Method | Decoding Time (s) | | | Line Coverage (%) | | |
|--------|---------|---------|---------|---------|---------|---------|
| | $n = 3$ | $n = 5$ | $n = 7$ | $n = 3$ | $n = 5$ | $n = 7$ |
| `DiffuCoder` | 12.2 | 19.5 | 26.8 | 91 | **94** | 94 |
| + EB-SAMPLER | 8.9 | 14.3 | 19.8 | 86 | 88 | 88 |
| + DIFFTESTER | **7.8** | **12.6** | **17.6** | **92** | 93 | **94** |

```
Without acceleration:                          With acceleration:

Step 1:                                        Step 1:
int main() {                                   int main() {
<int {                                         <int {
 solution grid Solution {{                      solution grid Solution {{
00 vector0 { {{,B =B1,),, {,,     0},          00 vector0 { {{,B =B1,),, {,,     0},
>> ,), , {}{{ ==Square);                       >> ,), , {}{{ ==Square);
```   00 ,11B       , } solution return      ```   00 ,11B       , } solution return
 {                                              {
,0 1 ,,,10,,,1``,{;                            ,0 1 ,,,10,,,1``,{;
 is  B,  largest11}                             is
}
int main() {                                   int main() {
    vector grid>>>> grid =11 vector grid01{{ 0     0 , 0,    01,,1 int},},,,00 { = 0,;      vector grid>>>> grid =11 vector grid01{{ 0     0 , 0,    01,,1 int},},,,00 { = 0,;
,,,10 1  }                                     ,,,10 1  }
 ',   , {                                       ',   , {
},}}1  .  ```11 ; the 1,{,,                    },}}1
}},``1  ,0 }

Step 11:                                        Step 11:
                                               int main() {
int main() {                                       Solution solution;
    Solution solution;                             vector<vector<int>> grid = {{1, 1, 1}, {1, 0, 1}, {1, 1, 1}};
    vector<vector<int grid grid111111,11,,1,11111111111111111111111111111111111111    assert(solution.largest1BorderedSquare(grid) == 9);
                                               }
int main() {                                   int main() {
    Solution solution;                             Solution solution;
    vector<vector<int grid grid =1111111111,11111111111111111111111111111111111111        vector<vector<int>> grid = {{1, 1, 1, 0},{1, 0, 1, 1},{1, 1, 1, 1},{1, 0, 0, 0}};
                                                   assert(solutionargestBBBorderedSquare(grid) == 9);
                                                   return

Step 21:                                        Step 21:
                                               int main() {
int main() {                                       Solution solution;
    Solution solution;                             vector<vector<int>> grid = {{1, 1, 1}, {1, 0, 1}, {1, 1, 1}};
    vector<vector<int>> grid =11111,1,111111111111111111111111111111BB              assert(solution.largest1BorderedSquare(grid) == 9);
                                               }
int main() {                                   int main() {
    Solution solution;                             Solution solution;
    vector<vector<int>> grid = {{1, ,, 1, 1}, 1, 1, 1}, 1, 1, 11 11 1111111B11          vector<vector<int>> grid = {{1, 1, 1, 0},{1, 0, 1, 1},{1, 1, 1, 1},{1, 1, 1, 1}};
                                               }
```

Figure 5: Case study on `Dream` illustrating the decoding process with and without DIFFTESTER.

DIFFTESTER, the line coverage remains almost identical to the baseline without any acceleration, while EB-SAMPLER leads to a significant drop in coverage. We attribute the superior performance of DIFFTESTER in terms of both efficiency and quality to its ability to exploit repetitive structural patterns that are inherent to the UTG task. This task-specific insight allows DIFFTESTER to accelerate generation without compromising the quality of the resulting test cases.

## 5.2 CASE STUDY

We present a concrete case in Figure 5 to further illustrate the impact of DIFFTESTER. For the same focal method, we compare the decoding process of `Dream` with and without DIFFTESTER. The case clearly shows that DIFFTESTER substantially accelerates UTG: within 21 decoding steps, the version with DIFFTESTER has already generated nearly complete and valid test cases, whereas the version without DIFFTESTER still fails to produce a proper test case and contains numerous syntax errors.

## 5.3 DISTRIBUTION OF INFERENCE STEPS WITH DIFFTESTER

In this section, we analyze the distribution of the number of inference steps after applying DIFFTESTER. The experiments are conducted on `DiffuCoder` across three benchmarks, with the predefined generation length set to 128. Under the default remasking strategy, the model decodes two tokens at each step, which results in 64 inference steps without DIFFTESTER. The distribution of inference steps after applying DIFFTESTER is shown in Figure 6. We observe that the number of steps follows an approximately normal distribution, with most cases concentrated at values significantly smaller than 64. Moreover, the number of inference steps adapts to the characteristics of each focal method, making the approach well-suited for diverse testing scenarios.

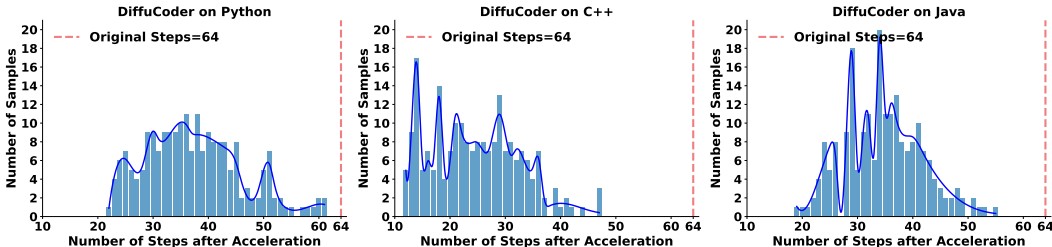

Figure 6: Distribution of the number of inference steps after applying DIFFTESTER on `DiffuCoder`. The red dashed line indicates the original fixed step number without acceleration.

## 6 RELATED WORK

### 6.1 UNIT TEST GENERATION

Unit tests (Olan, 2003; Runeson, 2006) are essential for verifying the behavior of program units and serve as a foundational mechanism for early fault detection and prevention in software development. However, writing unit tests manually is highly labor-intensive, which has motivated extensive research on automated unit test generation (Dakhel et al., 2024; Aniche, 2022). Traditional techniques primarily rely on search-based (Delgado-Pérez et al., 2022; McMinn, 2004), random-based (Pacheco et al., 2007), constraint-based (Csallner et al., 2008), and symbolic execution-based approaches (Baldoni et al., 2016), with widely used tools such as EvoSuite (Fraser & Arcuri, 2011), Pynguin (Lukasczyk & Fraser, 2022), and KLEE (Cadar et al., 2008). Nevertheless, these traditional approaches often suffer from limitations such as low coverage (Barr et al., 2014).

With the rise of large language models (LLMs) (Achiam et al., 2023), recent advances have increasingly explored their application to automated test case generation. Representative approaches include ChatTester (Yuan et al., 2024), TestPilot (Schäfer et al., 2023), and ChatUnitTest (Chen et al., 2024). Although LLMs have significantly improved test coverage, existing LLMs used for unit test generation almost exclusively adopt a token-by-token generation paradigm (Li et al., 2025a), which constrains the efficiency of the unit test generation.

### 6.2 DIFFUSION LARGE LANGUAGE MODEL

Diffusion language models have recently become a focal point in AI research (Sahoo et al., 2024; Liang et al., 2025) and can be categorized into continuous and discrete formulations (Li et al., 2025c). Empirical evidence (Zhao et al., 2025) suggests that discrete diffusion language models scale more effectively to larger model sizes, which has led to the development of diffusion large language models such as LLaDA (Nie et al., 2025b) and Gemini Diffusion (DeepMind, 2025). Within this line of work, MMaDA (Yang et al., 2025) and LLaDA-V (You et al., 2025) extend dLLMs to the multimodal domain; TreeDiff (Zeng et al., 2025) explores integrating dLLMs with the structural characteristics of code; DLLM-Cache (Liu et al., 2025b) and Sparse-dLLM (Song et al., 2025a) propose inference acceleration strategies based on KV-cache mechanisms; DiffuCoder (Gong et al., 2025) and LLaDA-1.5 (Zhu et al., 2025) investigate reinforcement learning tailored to dLLMs.

## 7 CONCLUSION

In this paper, we present an in-depth analysis of the pervasive repetitive patterns in unit test generation and the characteristics of dLLMs' inference. Building on this analysis, we propose DIFFTESTER, the first acceleration framework specifically designed for dLLMs in unit test generation. The central idea of DIFFTESTER is to extract repetitive patterns in intermediate unit test cases and leverage them to decode a larger number of tokens per inference step. Extensive experimental results demonstrate that DIFFTESTER can substantially accelerate dLLMs in unit test generation while preserving test coverage, and that it generalizes effectively across different dLLMs and programming languages. We believe that this work provides a practical and scalable solution for improving the efficiency of unit test generation in software development.

## 8 REPRODUCIBILITY STATEMENT

To ensure the reproducibility of our results, we have made all relevant code and datasets publicly available at https://anonymous.4open.science/r/DLM4UTG. The repository includes detailed instructions for setting up the environment, running experiments, and reproducing all results presented in the paper. We encourage the community to use these resources to verify our findings and to facilitate further research in this area.

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

## A    LLM USAGE STATEMENT

We used Large Language Models solely as a tool to assist with the linguistic aspects of our manuscript. Specifically, the model was employed to help with translation, refine grammar, enhance clarity, improve conciseness, and optimize word choice. At no point did the model contribute any new ideas, original content, or substantive changes to the manuscript. Its function was limited strictly to editing and language optimization, ensuring that the scientific integrity and originality of our work remained entirely unaffected by the use of the model.

## B    ETHICS STATEMENT

Our work focuses on improving software quality assurance and does not involve any foreseeable ethical risks. The benchmark used in our experiments is constructed entirely from publicly available sources, without the use of private or otherwise sensitive information. No human subjects, personally identifiable data, or sensitive content are involved in this research.

## C    ADDITIONAL DISCUSSION

### C.1    DETAILED SETUP

#### C.1.1    MODELS

We primarily conducted experiments using the dLLMs `DiffuCoder-7B-cpGRPO` (Gong et al., 2025) (`DiffuCoder` in this paper for short) and `Dream-v0-Instruct-7B` (Ye et al., 2025) (`Dream` in this paper for short).

**DiffuCoder-7B-cpGRPO** is a code generation dLLM developed by Apple. It is a refined variant of DiffuCoder-Instruct, further improved using Coupled-GRPO reinforcement learning. The model is initialized from DiffuCoder-7B-Instruct and post-trained on 21,000 code samples for one epoch.

**Dream-v0-Instruct-7B** is a variant of the Dream 7B model developed by the HKU NLP Group with 7 billion parameters. It is an instruction-tuned (SFT) large diffusion language model. It is trained on a mixture of text, math, and code data, leveraging weight initialization from auto-regressive models for more efficient learning, and it demonstrates strong performance on general, coding, and reasoning tasks.

#### C.1.2    METRICS

The following provides a detailed description of the metrics measured in our main experiments.

**Coverage.** A software testing metric indicating the percentage of source code lines (or branches) exercised by the generated test cases. Higher coverage generally means that the generated tests explore more of the code base, leading to better test quality. In our experiments, we use line coverage, calculated as the number of code lines covered by the test cases divided by the total number of code lines.

**Computational Cost (tflops).** The overall amount of computation required by the model during inference reflects the hardware resources needed to complete the task. In our experiments, we measure the average computational cost to generate a certain number of test cases consumed per problem across the entire benchmark.

**Time (s).** Time needed during inference to generate a certain number of test cases. In our experiments, we measure the average inference time consumed per problem across the entire benchmark.

**Throughput (tps).** The number of tokens generated per second (tokens per second, tps). It reflects how many outputs can be produced in parallel and is critical for large-scale deployment. In our experiments, throughput is calculated as the total number of tokens generated across the entire benchmark (excluding special tokens such as `[EOS]` and `[PAD]`) divided by the total GPU time used.

### C.1.3 HYPERPARAMETERS

Hyperparameters we used in the main experiments are listed in Table 3:

Table 3: Hyperparameters used in the main experiments.

| Model | Language | Steps | Temperature | Max length | Alg. temp.[1] | Threshold[2] | Step size[3] |
|-------|----------|-------|-------------|------------|-----------|-----------|-----------|
| DiffuCoder | Python | 64 | 1.5 | 128 | 0.2 | 0.02 | 2 |
|  | C++ | 64 | 1.5 | 192 | 0.2 | 0.02 | 2 |
|  | Java | 64 | 1.5 | 192 | 0.2 | 0.02 | 2 |
| Dream | Python | 64 | 1.0 | 128 | 0.2 | 0.02 | 2 |
|  | C++ | 64 | 1.0 | 192 | 0.2 | 0.02 | 2 |
|  | Java | 64 | 1.0 | 192 | 0.2 | 0.02 | 2 |

In our experiments with the EB-SAMPLER method, we adopted the parameters recommended in the original paper (Ben-Hamu et al., 2025). We used confidence as the *error proxy function*. And we used $\gamma = 0.1$.

### C.1.4 BENCHMARK

To evaluate the performance of our method, we experiment on the TestEval benchmark (Wang et al., 2025). TestEval is a benchmark for test case generation with LLMs. It includes 210 Python programs from an online programming platform, LeetCode. It has three metrics: overall coverage, targeted line/branch coverage, and targeted path coverage.

To further assess the generalizability of our method across multiple programming languages, we extend the TestEval benchmark by collecting corresponding C++ and Java implementations for the same set of 210 problems. These additional solutions are obtained from a publicly available GitHub repository licensed under the MIT license (see https://github.com/walkccc/LeetCode for details).

Building on TestEval, we also adapted the evaluation code for line coverage and branch coverage in Java and C++. For coverage measurement, we used pytest for Python, Maven for Java, and gcov for C++.

## C.2 MORE EXPERIMENTS

### C.2.1 COMPARISON WITH AR MODEL

In this section, we present a comparison between our method and an AR baseline model.

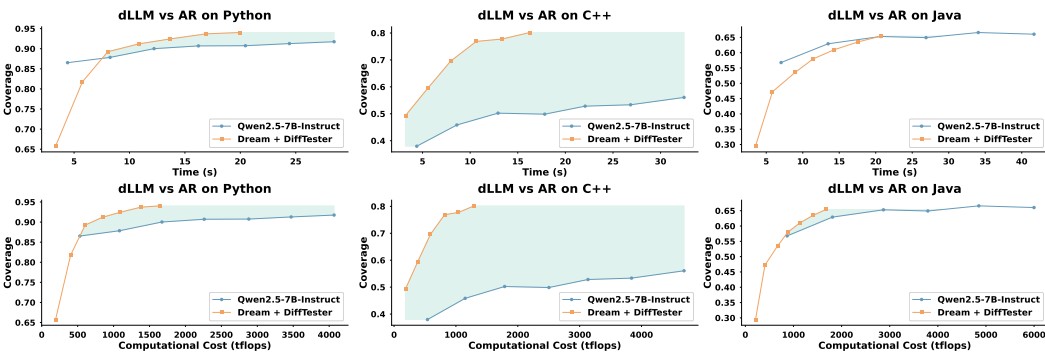

Figure 7: Dream + DIFFTESTER vs Qwen-2.5-7B-Instruct

From Figure 7, we can see that with our method, dLLMs require substantially less time and computational cost to achieve the same coverage in most cases compared to auto-regressive models of similar scale.

### C.2.2 MORE COVERAGE METRICS

In this section, we present some experiment results using branch coverage.

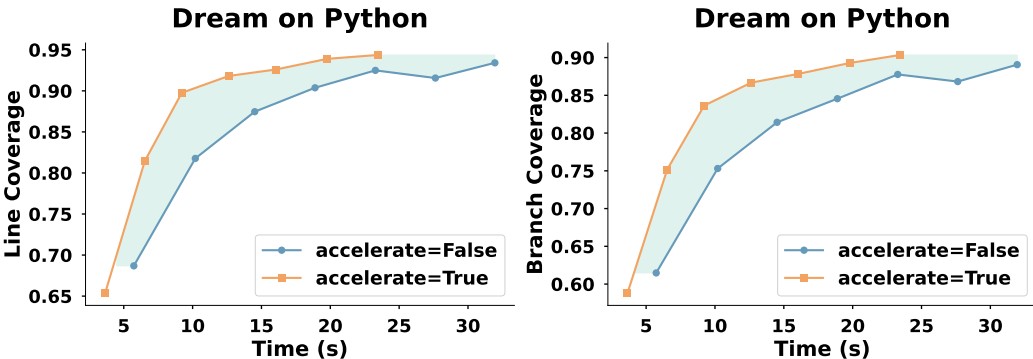

Figure 8: Comparison of line/branch coverage with and without DIFFTESTER at equal decoding time.

From Figure 8, we can see that using branch coverage as a metric yields results that are highly consistent with those of line coverage. That is the reason why we only reported line coverage in the main results.

### C.2.3 QUALITY VS SPEED

In this section, we demonstrate the syntactic correctness of the code generated with different dLLM generation speeds. Different speeds are caused by varying the number of tokens unmasked per step. The experiment is conducted on `DiffuCoder` model on Python language.

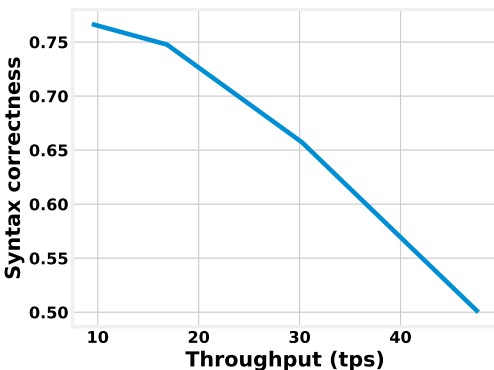

Figure 9: Syntactic correctness of the code generated with different generation speed.

From Figure 9, we can see that unmasking more tokens at each step can obviously improve speed, but it also degrades generation quality. The results highlight a clear trade-off between generation speed and syntactic correctness. As the number of tokens unmasked per step increases, the model is able to generate outputs at a substantially higher throughput. However, this acceleration comes at the cost of syntactic quality, as evidenced by the marked decline in syntax correctness.

### C.2.4 ABLATION STUDY ON THRESHOLD

During the dLLM denoising process, each unmasked token is assigned a confidence score, which reflects how certain the model is about its prediction. This confidence can be measured in two common ways: (1) as the probability of the selected token according to the model's output distribution, or (2) as the negative entropy of the token distribution, which captures the overall uncertainty across all

possible tokens. In all of our experiments, we simply use the first method, i.e., the probability of the selected token, to represent confidence.

In our method, only tokens whose confidence is greater than the specified threshold are unmasked. In this section, we show the influence of this threshold.

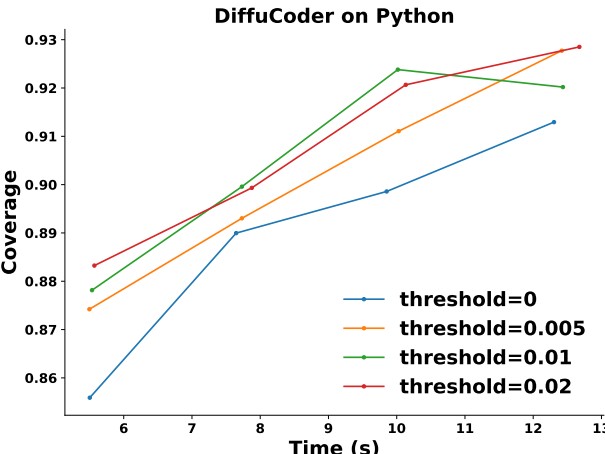

Figure 10: Coverage vs Time for different thresholds.

From Figure 10, we can know that accelerating without a threshold (threshold = 0) leads to a decrease in the final coverage. That is because the method potentially results in syntactically incorrect code, since it attempts to match even when syntax errors are present. A proper threshold (like 0.02) alleviates it.

### C.2.5 ABLATION STUDY ON STEP SIZE

We observed that applying the method at every denoising step is not necessary; using it only at certain steps has minimal impact on generation quality and the number of extra unmasked tokens, while significantly reducing the computational overhead introduced by the algorithm.

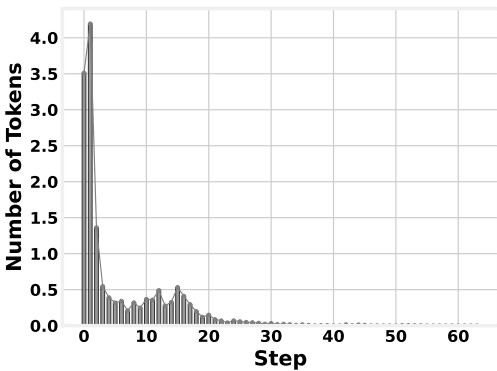

Figure 11: Number of Tokens Accelerated by Our Method for Every Step.

We conducted experiments utilizing both fixed-interval and dynamically adjusted step size strategies for applying our method. In the dynamically adjusted approach, the method is applied with greater frequency during the initial steps, while the interval between successive applications increases linearly in later steps. This design is motivated by our observation that the method is able to unmask a greater number of additional tokens during earlier steps compared to later ones, as illustrated in Figure 11. Empirically, we find that while the dynamically adjusted step size yields competitive results, a constant step size of 2 ultimately achieves the best overall performance.

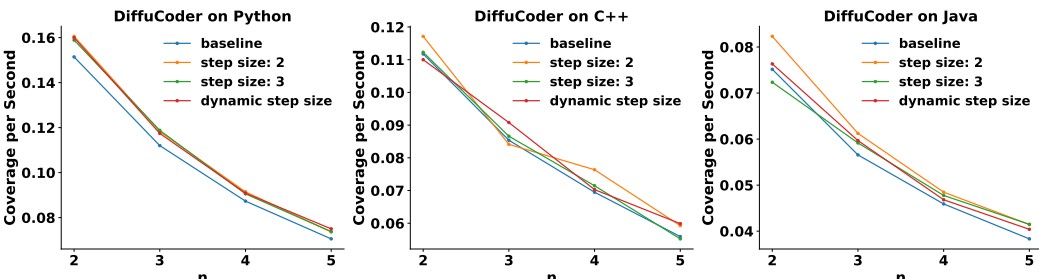

Figure 12: Results of different step sizes.

Figure 12 compares the results of different fixed step sizes and the dynamic step size strategy. The y-axis represents the average coverage divided by the average time, while the x-axis denotes the number of test cases generated simultaneously. The higher y-value means better performance. The curve labeled *baseline* is the case in which we apply our method every step. From the figure, we can know that not applying the method to every step produces better results. In general, a constant step size of 2 yields the best results.

### C.2.6 ABLATION STUDY ON [PAD] TOKEN

In this section, we examine the impact of a straightforward yet effective technique for further accelerating the generation process. During training, [PAD] tokens are appended to the ends of sequences to ensure uniform length, resulting in [PAD] tokens exclusively appearing at sequence termini in standard text. Building on this observation, we propose the following strategy: once a [MASK] token is decoded as a [PAD] token during inference, all subsequent positions in the sequence are immediately assigned as [PAD] tokens. This approach leverages the inherent structure of the training data to bypass unnecessary computations for positions that are, by construction, expected to be padding, thereby yielding additional improvements in generation efficiency.

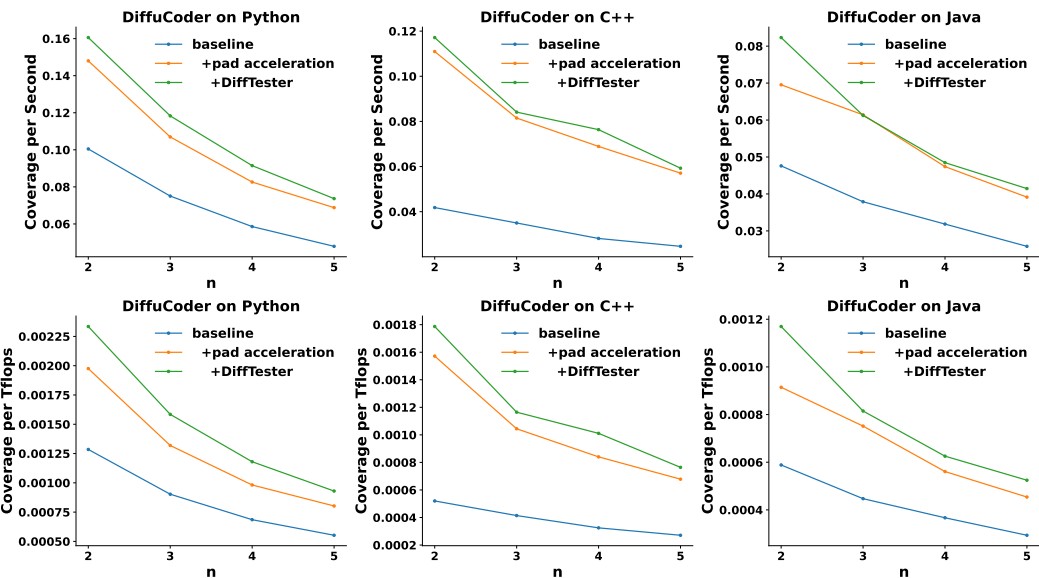

Figure 13: Acceleration result of pad acceleration and pattern acceleration.

Results are shown in Figure 13. The y-axis represents the average coverage divided by the average time or average computational cost, while the x-axis denotes the number of test cases generated simultaneously. The higher y-value means better performance. The curve labeled *baseline* means no

acceleration method applied. The curve labeled +*pad acceleration* means applying this trick on the baseline, while the curve labeled +DIFFTESTER means applying both this trick and our method.

From the results, we can see that this simple trick can also accelerate the generation process, while our AST-based pattern mining method further accelerates the process.

### C.3 PROMPTS

The system prompt template we used is as follows:

```
You are a professional who writes {language} testcases. You always
respond without any natural language descriptions. Especially,
your answer should contain only one testcase.
```

The prompt template we used in different programming languages is as follows:

**Python:**

```
Please write a test method for the function '{func_name}' given
the following program under test and function description. Your
answer should only contain one test input.

Program under test:
'''python
{program}
'''

Function description for '{func_name}':
'''txt
{description}
'''

Your testcase should begin with:
'''python
def test_{func_name}():
    solution = Solution()
'''
```

**Java:**

```
Please write a test method for the function '{func_name}' given
the following program under test and function description. Your
answer should only contain one test input.

Program under test:
'''java
{program}
'''

Function description for '{func_name}':
'''txt
{description}
'''

You can directly use `assertEquals` function for assertion. Your
testcase should be formatted as:
'''java
public class SolutionTest {
  @Test
  public void test_{func_name}() {
```

```
    }
}
```

**C++:**

```
Please write a testcase for the function '{func_name}' given the
following program under test and function description. Your answer
should only contain one test input.

Program under test:
```cpp
{program}
```

Function description for '{func_name}':
```txt
{description}
```

You can directly use `assert` function for assertion. Your
testcase should be formatted as:
```cpp
int main() {
    Solution solution;
}
```
```

### C.4 More Cases and Data Visualization

**Code is written on multiple lines :**

```java
public class SolutionTest {
  @Test
  public void test_longestCommonSubpath() {
    assert longestCommonSubpath(5, new int[][] {{1, 2, 3}, {1, 2, 3, 4}, {2, 3}}
      == 2;
  }
}
public class SolutionTest {
  @Test
  public void test_longestCommonSubpath() {
    int n = 3;
    int[][] paths = {{0,1,2}}
    int result = new Solution().longestCommonSubpath(n, paths);
    assertEquals(result, 2);
  }
}
```

**Overly complex data structures :**

```python
def test_pacificAtlantic():
    solution = Solution()
    heights = [[1,2,2,3,5],[3,2,3,4,3,4,3,2],[4,3,4,1,1],[5,1,1,2,4]]
    assert solution.pacificAtlantic(heights) == [[0,4],[1,3],[1,4],[2,2],[3,0],[3,1],[3,2],[4,0],[5,0]]
def test_pacificAtlantic():
    solution = Solution()
    heights = [[1,2,2,3,5],[3,3,3,4,4],[2,4,5,3,1],[6,7,1,4,5],[5,1,1,2,4]]
    expected = [[0,4], [1,3],[1,4],[3,0],[4,0]]
    assert solution.pacificAtlantic(heights)==expected
```

Figure 14: Examples of insignificant acceleration.

### C.4.1 ANALYSIS OF INSIGNIFICANT ACCELERATION

Figure 14 illustrates situations where the speedup effect is less pronounced. Pattern matching becomes challenging when code spans multiple lines. Furthermore, when the data structure is highly complex and contains a large number of variables that need to be decoded, the model still requires a large number of steps to complete the decoding process. These observations highlight the limitations of pattern-based speedup in scenarios involving multi-line code grammatical structures or complex data representations.

### C.4.2 MORE EXAMPLES OF STEP-BY-STEP COMPARISON GENERATION

Figure 15: Another step-by-step comparison example of acceleration and non-acceleration.

Figure 15 is another step-by-step comparison example of acceleration and non-acceleration. We clearly observe that, given the same number of decoding steps, our acceleration method produces a significantly greater number of decoded tokens compared to the unaccelerated baseline. This pronounced difference demonstrates that our method enables the generation of a more complete and diverse set of test cases within a shorter time frame, which directly contributes to higher test coverage. Therefore, this example also demonstrates that our method exhibits clear advantages in the domain of automated test case generation, providing robust technical support for improving both test coverage and software quality.

**DiffuCoder on Python :**

```python
def test_groupStrings():
    solution = Solution()
    assert solution.groupStrings(["a", "b", "ab", ""]) == [2, 2]

def test_groupStrings():
    solution = Solution()
    words1 = ["abc", "ac", "abcd", "cde", "ab"]
    assert solution.groupStrings(words1) == [2, 3]
```

Figure 16: Example of DiffuCoder on Python.

**DiffuCoder on Java :**

```java
public class SolutionTest {
  @Test
  public void test_countCompleteComponents() {
    int n = 6;
    int[][] edges = {{0, 1}, {1,2}, {3,4}};
    assertEquals(1, new Solution().countCompleteComponents(n, edges));
  }
}
public class SolutionTest {
  @Test
  public void test_countCompleteComponents() {
    int n = 6;
    int[][] edges = {{0, 1}, {1, 2}, {3, 4}, {4, 5}};
    assertEquals(2, Solution().countCompleteComponents(n, edges));
  }
}
```

Figure 17: Example of DiffuCoder on Java.

**DiffuCoder on C++ :**

```cpp
int main() {
    Solution solution;
    vector<vector<int>> heights = {{1},{2}};
    assert(solution.minimumEffortPath(heights) == 0);
    return 0;
}

int main() {
    Solution solution;
    vector<vector<int>> heights{{1,2,2},{3,8,2},{5,3,5}};
    assert (solution.minimumEffortPath(heights) == 2);
    return 0;
}
```

Figure 18: Example of DiffuCoder on C++.

**Dream on Python :**

```python
def test_longestCommonPrefix():
    solution = Solution()
    arr1 = [1234 , 123456]
    arr2 = [12345, 123456]
    assert solution.longestCommonPrefix(arr1, arr2) == 4

def test_longestCommonPrefix():
    solution = Solution()
    arr1 = [123, 456]
    arr2 = [1234, 45678]
    assert solution.longestCommonPrefix(arr1, arr2) == 3
```

Figure 19: Example of Dream on Python.

**Dream on Java :**

```java
public class SolutionTest {
  @Test
  public void test_maxStrength() {
    int[] nums = {1, -2, -3, 0};
    assertEquals(6, new Solution().maxStrength(nums));
  }
}
public class SolutionTest {
  @Test
  public void test_maxStrength() {
    var nums = new int[] { 3, -1, -5, 2, 4 };
    assertEquals(20, new Solution().maxStrength(nums));
  }
}
```

Figure 20: Example of Dream on Java.

**Dream on C++ :**

```cpp
int main() {
  Solution solution;
  vector<vector<int>> board1 = {{1, 0}, {0, 1}};
  assert(solution.movesToChessboard(board1) == 0);
}

int main() {
  Solution solution;
  vector<vector<int>> board = {{1, 0, 1, 0}, {0, 1, 0, 1}, {1, 0, 1, 0}};
  assert(solution.movesToChessboard(board) == 1);
}
```

Figure 21: Example of Dream on C++.

### C.4.3 GENERATED EXAMPLES OF DIFFERENT MODELS ON DIFFERENT LANGUAGE TASKS

Figure 16,17,18,19,20,21 are the generated examples of two models, `DiffuCoder` and `Dream`, on three language tasks–Python, Java and C++. The code segments highlighted in blue represent the repetitive patterns identified within the unit test cases. Notably, these repetitive patterns are consistently observed in each language, indicating their universality across diverse coding environments. This observation suggests that different dLLMs can effectively leverage these repetitive patterns to accelerate the decoding process. By exploiting the presence of such patterns, the models are able to generate test cases more efficiently. These more motivating examples underscore the potential for repetitive pattern utilization as a general mechanism for boosting decoding efficiency in unit test generation scenarios.

### C.5 LIMITATION

A principal limitation of DIFFTESTER is the absence of empirical investigations into the acceleration of UTG on repository-level codebases, which would be of greater practical significance. This omission is primarily attributable to the constrained maximum context lengths supported by existing dLLM architectures. In contrast, repository-scale UTG tasks, such as those exemplified by TestGenEval (Jain et al., 2025), inherently demand the modeling of substantially longer contexts that exceed these architectural limits.

Another limitation arises from the inherent characteristics of certain contemporary code generation models, such as DreamCoder (Xie et al., 2025). Our method was not applied to these models primarily because their generative behavior tends to produce substantial amounts of natural language output, even when explicitly prompted to focus on code generation. This results in a low proportion of code segments, thereby constraining the potential acceleration benefits achievable by our method.

## C.6 ALGORITHM

---

**Algorithm 1:** Algorithm to merge two ASTs

---

**Input:** Root nodes of two ASTs: $node_1$ and $node_2$
**Output:** Merged AST root node

1   $merged\_node \leftarrow$ empty node;
2   **if** $node_1.type = node_2.type$ and $node_1$ is not error node **then**
3     $merged\_node.type \leftarrow node_1.type$;
4     **foreach** $child_1$ and $child_2$ from $node_1.children$ and $node_2.children$ **do**
5       $merged \leftarrow$ recursively merged root node of $child_1$ and $child_2$;
6       **if** $merged$ is not empty node **then**
7         add $merged$ to $merged\_node.children$;

8   **return** $merged\_node$;

---

---

**Algorithm 2:** Unmasking extra tokens via repetitive pattern

---

**Input:** $codelines \leftarrow$ all lines of code
```
/* merged_list represents list of merged node and its source
   code lines                                                    */
```
1   $merged\_list \leftarrow$ empty list of (node, list) tuple;
2   **for** $i = 0$ **to** $len(codelines)$ **do**
3     $current\_node \leftarrow$ root node of AST of this code line;
```
   /* The variable found indicates whether a node has already
      been successfully merged with the AST root node of the
      corresponding line of code, with the merge result being
      non-empty.                                                 */
```
4     $found \leftarrow False$;
5     **foreach** $(node, lines) \in merged\_list$ **do**
6       $merged\_node \leftarrow$ merged node of $node$ and $current\_node$;
7       **if** $merged\_node$ is not empty node **then**
8         change $node$ into $merged\_node$;
9         add $i$ to $lines$;
10        $found \leftarrow True$;
11        break the loop;

12     **if** $found$ is False **then**
13       $merged\_list.append((current\_node, [i]))$;

14   **foreach** $(merged\_ast, lines) \in merged\_list$ **do**
```
   /* Only those greater than 1 indicate the presence of a
      repetitive pattern.                                        */
```
15     **if** $len(lines) > 1$ **then**
16       **foreach** $i \in lines$ **do**
17         unmask tokens in $codelines[i]$ that match the $merged\_ast$;

---

