# OpenReview forum: "DiffTester: Accelerating Unit Test Generation for Diffusion LLMs via Repetitive Pattern"
_ICLR.cc/2026/Conference — ICLR 2026 Conference Withdrawn Submission_

### Official Review · Reviewer_86Xk · 2025-10-28

**Soundness:** 2
**Presentation:** 3
**Contribution:** 2
**Rating:** 4
**Confidence:** 4

**Summary:**

This paper proposes to explore diffusion LLMs in unit test generation. In particular, the paper proposes DiffTester, an acceleration framework to speed up the decoding time of similar unit tests for the same focal methods. The experimental results show that the proposed approach could outperform existing baselines in generating higher-coverage tests within given time budgets across three different programming languages.

**Strengths:**

The evaluation includes multiple programming languages.

The paper studies how diffusion LLMs could accelerate unit test code generation for the first time.

**Weaknesses:**

The current evaluation cannot support the usefulness of the proposed approach in practice. Although the paper claims that the proposed approach could achieve higher coverage within the same budget (as shown in Figure-4), the conclusion is not convincing as the curves do not converge. In other words, the given time budget is not sufficient enough to show the final stable status of the proposed approach and baselines. In fact, the paper only gives 25 seconds to compare the proposed approach and baselines. However, if authors check recent LLM-driven unit test generation techniques (e.g., CoverUP[a]), people often give sufficient time (i.e., spending hours or dozens of minutes) to run test generation techniques for one  real-worlds software repository, where the coverage curves become stable within the time budgets. In addition, it also leaves another question that, when there allows more sufficient time budget, whether the proposed approach could result in higher coverage compared baselines. The current outperformance within the limited time budget cannot support the practical usage of the proposed approach.

[a] CoverUp: Effective High Coverage Test Generation for Python. FSE’25.


In addition, although the proposed approach achieves higher coverage within a limited time budget, the coverage improvement does not come from tackling the key challenge in test generation. In fact, it is as expected that test methods for the same focal methods share the same structures (except the test inputs and oracles); but the critical challenge in test generation is how to generate valid and diverse test inputs (that could achieve the deep and hard-to-cover paths of the software), which is obviously not the focus of the work. Therefore, the significance of the proposed approach (i.e., how the work could benefit the field) needs more justification.


There are some unclear approach and experiment details in the current version. In particular, the complete process of extracting similar AST nodes is unclear. Is it performed in an iterative way along with the generation of more unit tests?  Moreover, would unit tests for different focal method share similar structures? In other words, can the approach only be applied to accelerate generating unit tests for the same focal methods OR for different focal methods?

**Questions:**

1.	Could authors justify the significance of the proposed approach by showing how the technique can achieve higher coverage when given sufficient time budgets?

2.	What are the overheads of program analysis part in the approach (i.e., common nodes extraction)?


3.	What is the complete process of extracting similar AST nodes?

---

> ### Author Response · Authors · 2025-11-23
> **Response to Reviewer 86Xk**
>
> Thank you for acknowledging the strengths of our work. We sincerely appreciate your feedback. We now present a detailed response to address your concerns and comments.
>
> `Weakness 1: Usefulness in Practice`
>
> We appreciate your concern about the usefulness of our work in practice. Let us address your concerns. Using our method does not reduce the final peak coverage given sufficient time, but it significantly decreases the time and computational resources required to reach that peak. In practical scenarios, where there are a large number of focal methods to test, this can greatly reduce both time and computational costs. Achieving peak coverage in much less time can therefore provide substantial practical benefits to the field.
>
> `Weakness 2: Significance of the Proposed Approach`
>
> Thank you for highlighting the input diversity challenge in UTG. We agree that generating valid and diverse test inputs is a critical issue, and that it is not the focus of our work. In real-world software, there are often numerous focal methods, and generating unit tests for all of them can be very time-consuming. Therefore, we argue that reducing the time required to reach peak coverage is just as important as achieving a higher peak coverage. Our method significantly decreases the time needed to reach peak coverage, which we believe constitutes a meaningful contribution to the field.
>
> `Weakness 3: Unclear Details`
>
> We apologize for any confusion regarding the process of extracting similar AST nodes. Here, we provide a detailed description of how to extract similar nodes given two ASTs:
>
> 1. **Root node check:** If the type of the root node of AST1 matches that of AST2, construct a new AST consisting of a single node with that shared type. Otherwise, return an empty AST.
> 2. **Recursive merging:** For each child of the root node in AST1 and each child of the root node in AST2, if the types match, recursively merge the corresponding subtrees. The root of each merged subtree becomes a child of the new AST.
> 3. **Return the result:** The newly constructed AST contains exactly "the similar nodes" we wanted.
>
> A pseudocode version of this procedure is also provided as Algorithm 1 in Appendix C.6.
>
> We believe that unit tests for different focal methods also share structural similarities, although these are less pronounced than those for a single focal method. In other words, our method can be applied to accelerate the generation of unit tests across different focal methods, but its effectiveness may not be as strong.
>
> `Question 1: Coverage Given Sufficient Time Budgets`
>
> Thank you for emphasizing the importance of achieving high coverage given sufficient time. Please see our response to `Weakness 1` for detailed discussion.
>
> `Question 2: Overheads of Program Analysis Part`
>
> We appreciate your concerns about the overhead of the AST operations part. In fact, dLLM inference accounts for the majority of the time consumed, while AST operations consume relatively little time. In our experiments, the inference time of the dLLM is roughly 10 times greater than the cost of the AST operations.
>
> `Question 3: Complete Process of Extracting Similar AST Nodes`
>
> Thank you for pointing out your confusion about the process of extracting similar AST nodes. Please see our response to `Weakness 3` for details.

---

### Official Review · Reviewer_jBu6 · 2025-10-29

**Soundness:** 2
**Presentation:** 3
**Contribution:** 2
**Rating:** 2
**Confidence:** 4

**Summary:**

This paper presents DiffTester, a specialized acceleration method for diffusion large language models (dLLMs) in unit test generation (UTG). DiffTester is designed based on one key assumption that unit tests for the same focal function share a very similar structure, having repetitive patterns. Based on this assumption, DiffTester identifies such patterns in the abstract syntax trees (ASTs) of the generated assertion statements, then merges these ASTs and fills in the common tokens to accelerate the diffusion process. Experimented with two dLLMs as baselines on a LeetCode-based test generation benchmark (TestEval), the authors demonstrate DiffTester reduces computational cost (tflops) and decoding time (seconds, s), increases throughput (tokens per second, tps), while maintaining the line coverage of the generated test cases. The authors also extend TestEval to Java and C++ to show the generalizability of DiffTester.

**Strengths:**

1. Under the same time budget, DiffTester achieves higher line coverage than the baseline models, and eventually reaches the same maximum achievable coverage.

2. Comprehensive analysis: the authors not only compare to the baseline dLLMs without DiffTester acceleration, but also compare to general purpose acceleration methods for dLLMs, and demonstrate advantages in test generation.

**Weaknesses:**

1. False statement/assumption. On line 45–47, the authors cite Yang et al. [1] for their motivation to use dLLMs for UTG: the next-token prediction nature of LLMs, which increases time consumption and computational cost, is the reasoning for inefficient UTG. If I read and understand their paper correctly, Yang et al.’s main conclusion is that the reason for inefficient UTG is the LLM-generated tests being invalid, and thus failing to achieve high coverage and detect real bugs on the real-world defect dataset Defects4J. However, dLLMs cannot help in such a case, or at least not demonstrated by the authors.

2. The authors further mentioned that the quality of generated UT is inversely proportional to the number of tokens generated per step in a dLLM’s inference process (line 56), which motivates the authors to study an acceleration method for this specific use case. However, generating multiple tokens per step, where time consumption and computational cost are lower than LLMs, is the whole motivation that the authors argue for dLLM for UTG. This is a circular argument.

3. The identified common structure is limited to simple algorithm/data structure contest problem code, but may not generalize to real-world software. Real-world software contains many more functions, and unit tests for large-scale software often require a function-call sequence, which invokes multiple functions/methods/APIs under the same class/module to set up or initialize the focal function [2,3]. In this paper, the authors oversimplify this to the same assertion structure and let the model provide inputs and outputs.

4. If the authors’ assumption and argument hold, the problem of UTG falls back to fuzzing with LibFuzzer. The authors assumed that the tests have the same structure, and argued for throughput and cost as the limiting factor in UTG.

   ```c
   // fuzz_target.cc
   extern "C" int LLVMFuzzerTestOneInput(const uint8_t *Data, size_t Size) {
     DoSomethingInterestingWithMyAPI(Data, Size);
     return 0;  // Values other than 0 and -1 are reserved for future use.
   }
   ```

   LibFuzzer achieves the exact purpose of DiffTester, offering a shared structure (the fuzz target), and varying the input data. It has much higher throughput and lower computation cost than any LLM or dLLM. All you need is to let the LLM generate a fuzz target.

5. In the experiment on coverage vs. time (Figure 4), the authors could also include a comparison with a 7B LLM for reference.

6. The authors conclude that DiffTester does not compromise the maximum achievable test coverage. However, only line coverage is reported, and the other more fine-grained metrics like branch coverage and path coverage in TestEval are missing. Considering that the benchmark is constructed from LeetCode, where the solutions are usually short in lines, line coverage, particularly the average line coverage across all submissions, is a poor metric to indicate any performance difference. I suggest the authors not justify or argue for anything based on this result. Instead, I suggest the authors evaluate the method on real-world test generation benchmarks like TestGenEval [4].



[1]: Yang et al., "On the Evaluation of Large Language Models in Unit Test
Generation" ASE 2024.

[2]: Pacheco et al., "Feedback-directed Random Test Generation" ICSE 2007.

[3]: Lyu et al., "Prompt Fuzzing for Fuzz Driver Generation" CCS 2024.

[4]: Jain et al., "TestGenEval: A Real World Unit Test Generation and Test Completion Benchmark" ICLR 2025.

**Questions:**

Please address concerns in **Weaknesses**.

---

> ### Author Response · Authors · 2025-11-23
> **Response to Reviewer jBu6 (1/2)**
>
> Thank you for acknowledging the strengths of our work. We sincerely appreciate your feedback. We now present a detailed response to address your concerns and comments.
>
> `Weakness 1: False statement/assumption`
>
> Thank you for pointing out this issue. We cite the paper by Yang et al. not for its main conclusion about test invalidity, but because it identifies several sources of inefficiency in UTG. While dLLMs can indeed generate tokens in parallel and are therefore theoretically more efficient than auto-regressive models. We apologize if our citation caused any confusion.
>
> `Weakness 2: Circular argument`
>
> We appreciate your concern regarding our motivation. Indeed, we argue for using dLLMs in UTG because they can generate multiple tokens per step. However, although dLLMs can decode multiple tokens in a single step, simply increasing the number of tokens decoded per step typically leads to a degradation in generation quality (e.g., decreased coverage). Our work addresses this trade-off with a novel method that increases the number decoded per step **without sacrificing output quality**.
>
> `Weakness 3: Oversimplifying to the same assertion structure`
>
> Thank you for highlighting the difference between real-world test cases and ours. Below is an example of a real-world test case (originally from https://github.com/django/django/blob/main/tests/migrations/test_executor.py#L906-L959). It contains multiple functions and API calls rather than a repeated assertion structure. It is also clear that the test case includes many recurring patterns, such as invoking graph.add_node and constructing FakeMigration objects. These similar structures can be effectively detected by our algorithm and used to accelerate the generation process. We hope this example helps illustrate that our method generalizes beyond simple assertion patterns.
>
> ```python
> def test_minimize_rollbacks_branchy(self):
>     r"""
>     Minimize rollbacks when target has multiple in-app children.
>
>     a: 1 <---- 3 <--\
>             \ \- 2 <--- 4
>             \       \
>     b:      \- 1 <--- 2
>     """
>     a1_impl = FakeMigration('a1')
>     a1 = ('a', '1')
>     a2_impl = FakeMigration('a2')
>     a2 = ('a', '2')
>     a3_impl = FakeMigration('a3')
>     a3 = ('a', '3')
>     a4_impl = FakeMigration('a4')
>     a4 = ('a', '4')
>     b1_impl = FakeMigration('b1')
>     b1 = ('b', '1')
>     b2_impl = FakeMigration('b2')
>     b2 = ('b', '2')
>     graph = MigrationGraph()
>     graph.add_node(a1, a1_impl)
>     graph.add_node(a2, a2_impl)
>     graph.add_node(a3, a3_impl)
>     graph.add_node(a4, a4_impl)
>     graph.add_node(b1, b1_impl)
>     graph.add_node(b2, b2_impl)
>     graph.add_dependency(None, a2, a1)
>     graph.add_dependency(None, a3, a1)
>     graph.add_dependency(None, a4, a2)
>     graph.add_dependency(None, a4, a3)
>     graph.add_dependency(None, b2, b1)
>     graph.add_dependency(None, b1, a1)
>     graph.add_dependency(None, b2, a2)
>
>     executor = MigrationExecutor(None)
>     executor.loader = FakeLoader(graph, {
>         a1: a1_impl,
>         b1: b1_impl,
>         a2: a2_impl,
>         b2: b2_impl,
>         a3: a3_impl,
>         a4: a4_impl,
>     })
>
>     plan = executor.migration_plan({a1})
>
>     should_be_rolled_back = [b2_impl, a4_impl, a2_impl, a3_impl]
>     exp = [(m, True) for m in should_be_rolled_back]
>     self.assertEqual(plan, exp)
> ```
>
> `Weakness 4: Fall back to fuzzing with LibFuzzer`
>
> We appreciate the comment that UTG may resemble fuzzing with LibFuzzer. In fact, our method can accelerate not only the generation of similar fuzz-target structures, but also other parts of the code, such as the construction of complex inputs. As illustrated in the example above, the constructed input data is highly complex and contains many API calls. For such cases, it is nearly impossible to devise a template—either manually or with an LLM—that specifies which APIs to invoke and how many times they should be invoked. But our method can detect these similar structures and accelerate the generation process. In summary, our method goes beyond what LibFuzzer can handle, because many portions of the code cannot be abstracted into a fixed template.
>
> `Weakness 5: Comparison with 7B AR LLM`
>
> We appreciate your suggestion to include a 7B AR LLM baseline. In our responses to all reviewers, we have provided both explanations and experimental results comparing our method with an AR baseline. The results show the efficiency improvement compared with AR LLM in terms of generation speed and computational efficiency. We invite you to refer to those results and hope they can address your concern.

---

> ### Author Response · Authors · 2025-11-23
> **Response to Reviewer jBu6 (2/2)**
>
> `Weakness 6: More metrics and real-world UTG benchmarks`
>
> **More metrics**: We appreciate your suggestion to use additional metrics and real-world benchmarks. In fact, we evaluated more than just line coverage. Based on our experiments, other coverage metrics yield results that are highly consistent with those from line coverage. Therefore, we only report line coverage in the manuscript. For example, here are some results based on branch coverage. In the table, each cell shows **average time** (s) / **line coverage** / **branch coverage**, respectively. We also provide a figure illustrating these results in the new version of our paper in Appendix C.2.2.
>
> | n | Dream | Dream + DIFFTESTER |
> |:-:|:-----:|:------------------:|
> | 1 | 5.73 / 0.69 / 0.61 | 3.64 / 0.65 / 0.59 |
> | 2 | 10.20 / 0.82 / 0.75 | 6.53 / 0.81 / 0.75 |
> | 3 | 14.51 / 0.87 / 0.81 | 9.21 / 0.90 / 0.84 |
> | 4 | 18.89 / 0.90 / 0.85 | 12.63 / 0.92 / 0.87 |
> | 5 | 23.27 / 0.92 / 0.88 | 16.03 / 0.93 / 0.88 |
> | 6 | 27.63 / 0.92 / 0.87 | 19.78 / 0.94 / 0.89 |
>
> **Real-world benchmarks**: Although dLLMs show promising potential, they are still in the early stages of development. Their fundamental capabilities are not yet sufficient for generating unit tests on repository-level benchmarks. For example, existing open-source dLLMs have relatively small context windows, while repo-level coding tasks typically require much longer contextual inputs. In our preliminary attempts on the TestGenEval dataset, we observed that current dLLMs have difficulty producing plausible test cases under such settings. Therefore, we did not include TestGenEval results in the manuscript. We also mentioned this in our manuscript in Appendix C.5. However, as an increasing number of researchers are now focusing on dLLMs, recent works have continuously improved their capabilities. We believe that as the fundamental abilities of dLLMs continue to advance, our method will become directly applicable to stronger future dLLMs and can be effectively used for repository-level UTG. The example we demonstrated in the response to `Weakness 3` shows that our method has great potential to accelerate real-world repository-level UTG.

---

> > ### Comment · Reviewer_jBu6 · 2025-11-26
> >
> > I would like to thank the authors for their rebuttal. After reading the rebuttal and other reviews, I acknowledge the main motivation of this paper: improving the efficiency of dLLMs in test generation without sacrificing test quality. dLLMs show some potential for their inference efficiency; however, whether they are useful for test generation is unclear. Thanks to the authors' effort in finding the real-world example in Django. However, as we can see in the same file, only the last two test functions exhibit such a pattern out of 1000 lines. Thanks for providing the new table, which offers more insight.
> >
> > In general, I think the author should further justify the necessity of dLLM in this specific use case, and thus, it is worth improving their efficiency.

---

> > > ### Author Response · Authors · 2025-11-26
> > > **Response to Reviewer jBu6**
> > >
> > > Thank you for acknowledging the main motivation of our work. We appreciate your feedback. We now present a detailed response to further address your concerns.
> > >
> > > `Repetitive Structures Beyond the Provided Example`
> > >
> > > In the same Django test file from which our example is taken, repeated patterns are not limited to the specific snippet we showcased. Other test functions also contain various forms of structural repetition, though to different extents. For instance, APIs related to `self.assert` and multiple `executor` APIs are invoked hundreds of times across the file. In addition, numerous instances of data constructions follow highly similar formats. Although these patterns may be less visually striking than the example we highlighted, they can still be effectively captured and leveraged by our method to accelerate the generation process.
> > >
> > > `Necessity of dLLM`
> > >
> > > In recent years, LLMs have significantly improved coding efficiency, and practices such as “vibe coding” have made machine-generated code increasingly common in real-world development. As a result, the demand for automatically generating unit tests has grown substantially, making the efficiency of UTG more crucial than ever.
> > >
> > > DLLMs inherently exhibit higher generation efficiency than auto-regressive LLMs due to their unique advantage of parallel decoding. Although current dLLMs are still in an early stage, the dLLM research community is rapidly expanding, and these models are improving at a fast pace. Recent dLLMs like Dream-Coder [1] have achieved code generation capabilities comparable to those of AR LLMs. We believe that dLLMs will become increasingly powerful. They will play an increasingly important role in test generation.
> > >
> > > Given this trend, we argue that investigating how to accelerate UTG with dLLMs is both timely and valuable. As dLLMs continue to mature, methods like ours will become increasingly important for practical adoption.
> > >
> > > [1]: Zhihui Xie et al. Dream-coder7b, 2025. URL https://hkunlp.github.io/blog/2025/dream-coder

---

### Official Review · Reviewer_GEZX · 2025-10-29

**Soundness:** 3
**Presentation:** 3
**Contribution:** 2
**Rating:** 4
**Confidence:** 4

**Summary:**

The paper’s key strength is a clever, diffusion-style formulation for unit-test generation that leverages parallel, iterative denoising to mine and lock in shared structural patterns across a batch—yielding impressive throughput and coverage gains. Empirically, results look strong and suggest real practicality for generating many tests per focal method. However, the work underexplores the reliability of generated assertions (e.g., semantic correctness, flakiness, and fault-revealing power) and lacks tighter, apples-to-apples comparisons with established LLM-based UTG baselines under the same budget. Suitable comparators with similar input/output include hybrid and prompt-engineered autoregressive approaches such as CODAMOSA, large-scale evaluations of LLMs for Java/Python JUnit/pytest generation, and recent comparative studies that analyze correctness and coverage—against which a direct head-to-head would clarify how much of the lift comes from diffusion decoding versus other factors.

**Strengths:**

1. Clever use of diffusion for UTG.

Iterative mask→predict→remask exposes mid-step drafts across a batch, letting you mine shared structure and lock it in—an original angle for speeding up test generation.

2, Impressive empirical gains.

Shows higher throughput/coverage at similar budgets (parallel per-step decoding + pattern guidance), suggesting real practicality for large batches of tests.

**Weaknesses:**

1. Assertion reliability under-analyzed.

The classic “oracle” issue remains: are the asserts semantically correct, flaky-free, and non-overfitted? Little on mutation/fault-revealing power, metamorphic checks, or human vetting rates.

2. Limited apples-to-apples baselines.

Needs stronger head-to-head with state-of-the-art LLM UTG systems under the same time/compute budget, including hybrid SBST+LLM and prompt-engineered AR LLMs.

* CODAMOSA: Escaping Coverage Plateaus in Test Generation with Pre-trained Large Language Models (ICSE 23)

More baselines are available at: LLMs and Prompting for Unit Test Generation: A Large-Scale Evaluation (ASE 24) regarding the coverage and effectiveness

**Questions:**

I agree that tests are very duplicated. But as for coverage, traditional software testing tools (Codemosa for Python and EvoSuite for Java) can do well. How your approach is compared with those approaches?

---

> ### Author Response · Authors · 2025-11-23
> **Response to Reviewer GEZX**
>
> Thank you for acknowledging the strengths of our work. We sincerely appreciate your feedback. We now present a detailed response to address your concerns and comments.
>
> `Weakness 1: Assertion reliability under-analyzed`
>
> We appreciate your concern regarding the "test oracle" issue. We agree that this is a critical aspect in UTG. However, the focus of our work is on accelerating UTG without compromising generation quality. Addressing the "test oracle" issue to improve generation quality is therefore beyond the scope and purpose of this paper. We have emphasized this point in our responses to all reviewers and hope this clarifies our position.
>
> `Weakness 2: Limited apples-to-apples baselines`
>
> We appreciate your suggestion to include auto-regressive model baselines. In our responses to all reviewers, we have provided both explanations and experimental results comparing our method with an AR baseline. The results show the efficiency improvement compared with AR LLM in terms of generation speed and computational efficiency. We invite you to refer to those results to better understand our thoughts.
>
> `Question: Comparing with traditional software testing tools`
>
> We acknowledge your concern about comparing our method with traditional software testing tools. The primary objective of our work is to accelerate UTG rather than to produce higher-quality unit tests. Quality-enhancing tools like Codamosa or EvoSuite can likewise be incorporated to improve the test cases generated by dLLMs, but that is beyond the scope of our work.

---

### Official Review · Reviewer_v4VJ · 2025-11-02

**Soundness:** 2
**Presentation:** 3
**Contribution:** 2
**Rating:** 6
**Confidence:** 4

**Summary:**

This paper proposes DiffTester, an acceleration framework for diffusion Large Language Models (dLLMs) in unit test generation. The key insight is that unit tests for the same focal method often share repetitive structural patterns, which can be exploited to generate multiple tokens per inference step. The authors use Abstract Syntax Tree (AST) analysis to identify common patterns across test cases and adaptively increase tokens generated per step. Experiments on TestEval benchmarks across Python, Java, and C++ show 1.5-2.8× speedups while maintaining test coverage.

**Strengths:**

- Well-Motivated Observation: The observation that unit tests exhibit repetitive structural patterns (Section 2.2, Figure 1) is well-demonstrated. The use of AST-based pattern extraction is a natural and principled approach to leverage this observation.
- Training-Free Approach: DiffTester requires no model retraining or fine-tuning, making it immediately applicable to existing dLLMs
- Comprehensive Multi-Language Evaluation: The extension of TestEval to C++ and Java (TestEval-C++ and TestEval-Java) strengthens the evaluation and demonstrates generalization across programming languages. The observation that C++ shows greater acceleration (2.45× vs 1.55× for Python) due to more structural repetition is valuable

**Weaknesses:**

- Limited Baseline Comparisons: The paper only compares against one general dLLM acceleration method (EB-Sampler). Comparisons with other acceleration techniques or autoregressive models with speculative decoding would strengthen the evaluation. What about comparing with standard autoregressive LLMs using techniques like speculative decoding?
- Scalability Concerns: The evaluation is limited to relatively simple, single-function benchmarks (210 LeetCode problems). Real-world unit test generation often involves complex dependencies, class hierarchies, and repository-level context
- Generalization Beyond UTG: While the method is specifically designed for unit test generation, could similar pattern-based acceleration apply to other code generation tasks (e.g., documentation generation, code translation)?

**Questions:**

- Figure 4 Discrepancy: Why do settings with higher time budgets sometimes exhibit worse performance than those with lower time budgets in Figure 4?
- Method Acceleration Analysis: Could you provide a theoretical or empirical analysis outlining the conditions under which this method offers significant versus minimal acceleration?
- Computational Cost Breakdown: What is the breakdown of computational costs between dLLM inference and AST operations?
- Performance Comparison: How does the performance compare to autoregressive LLMs, and what is the performance gain relative to speculative decoding for autoregressive LLMs?

---

> ### Author Response · Authors · 2025-11-23
> **Response to Reviewer v4VJ (1/2)**
>
> Thank you for acknowledging the strengths of our work. We sincerely appreciate your feedback. We now present a detailed response to address your concerns and comments.
>
> `Weakness 1: Limited Baseline Comparisons`
>
> We appreciate your suggestion to include auto-regressive model baselines. In our response to all reviewers, we have provided both explanations and experimental results comparing our method with an AR baseline. The results show the efficiency improvement compared with AR LLM in terms of generation speed and computational efficiency. We invite you to refer to those results and hope they can address your concern.
>
> `Weakness 2: Scalability Concerns`
>
> Thank you for highlighting the scalability concerns. Although dLLMs show promising potential, they are still in the early stages of development. Their fundamental capabilities are not yet sufficient for generating unit tests on repository-level benchmarks. For example, existing open-source dLLMs have relatively small context windows, while repo-level coding tasks typically require much longer contextual inputs. In our preliminary attempts on the TestGenEval dataset, we observed that current dLLMs have difficulty producing plausible test cases under such settings. Therefore, we did not include TestGenEval results in the manuscript. We also mentioned this in our manuscript in Appendix C.5.
>
> However, we note that an increasing number of researchers are now focusing on dLLMs, and recent works have continuously improved their capabilities. We believe that as the fundamental abilities of dLLMs continue to advance, our method will become directly applicable to stronger future dLLMs and can be effectively used for repository-level UTG. Below is an example of unit tests from the TestGenEval dataset (originally from https://github.com/django/django/blob/main/tests/migrations/test_executor.py#L906-L959), which clearly also exhibit repetitive structural patterns. These patterns can also be used to accelerate the generation process in theory, although we can not verify this in experiments because of the limited context length of current dLLMs.
>
> ```python
> def test_minimize_rollbacks_branchy(self):
>     r"""
>     Minimize rollbacks when target has multiple in-app children.
>
>     a: 1 <---- 3 <--\
>             \ \- 2 <--- 4
>             \       \
>     b:      \- 1 <--- 2
>     """
>     a1_impl = FakeMigration('a1')
>     a1 = ('a', '1')
>     a2_impl = FakeMigration('a2')
>     a2 = ('a', '2')
>     a3_impl = FakeMigration('a3')
>     a3 = ('a', '3')
>     a4_impl = FakeMigration('a4')
>     a4 = ('a', '4')
>     b1_impl = FakeMigration('b1')
>     b1 = ('b', '1')
>     b2_impl = FakeMigration('b2')
>     b2 = ('b', '2')
>     graph = MigrationGraph()
>     graph.add_node(a1, a1_impl)
>     graph.add_node(a2, a2_impl)
>     graph.add_node(a3, a3_impl)
>     graph.add_node(a4, a4_impl)
>     graph.add_node(b1, b1_impl)
>     graph.add_node(b2, b2_impl)
>     graph.add_dependency(None, a2, a1)
>     graph.add_dependency(None, a3, a1)
>     graph.add_dependency(None, a4, a2)
>     graph.add_dependency(None, a4, a3)
>     graph.add_dependency(None, b2, b1)
>     graph.add_dependency(None, b1, a1)
>     graph.add_dependency(None, b2, a2)
>
>     executor = MigrationExecutor(None)
>     executor.loader = FakeLoader(graph, {
>         a1: a1_impl,
>         b1: b1_impl,
>         a2: a2_impl,
>         b2: b2_impl,
>         a3: a3_impl,
>         a4: a4_impl,
>     })
>
>     plan = executor.migration_plan({a1})
>
>     should_be_rolled_back = [b2_impl, a4_impl, a2_impl, a3_impl]
>     exp = [(m, True) for m in should_be_rolled_back]
>     self.assertEqual(plan, exp)
> ```
>
> `Weakness 3: Generalization Beyond UTG`
>
> We appreciate your idea of generalizing our method beyond UTG. Both the documentation generation task and code translation task exhibit some repetitive structural patterns. Docstrings usually have a similar structure. The translated code also usually has a similar structure to the original code. These repetitions can also be exploited by our method for acceleration. However, since our work specifically focuses on UTG, we may extend our analysis to other tasks in the future.
>
> `Question 1: Figure 4 Discrepancy`
>
> We appreciate this observation. The cases where coverage slightly decreases as the number of test cases increases are outliers, and the magnitude of the decrease is very small, which does not affect the conclusions drawn from our results. As the number of test cases grows, the final coverage tends to stabilize; however, the randomness introduced by the sampling temperature can cause small fluctuations in the measured coverage, which explains the observed phenomenon.

---

> ### Author Response · Authors · 2025-11-23
> **Response to Reviewer v4VJ (2/2)**
>
> `Question 2: Method Acceleration Analysis`
>
> Thank you for this question. When the generated code contains a lot of literals, like the example below, our method will offer less significant acceleration. The reason is that we do not unmask literal tokens in advance. We consider these literals crucial for covering different code lines and branches, and therefore, we intentionally avoid interfering with their masking behavior. (See Section 3.2, line 235-240)
>
> ```python
> def test_pacificAtlantic():
>     solution = Solution()
>     heights = [[1,2,2,3,5],[3,2,3,4,3,4,3,2],[4,3,4,1,1],[5,1,1,2,4]]
>     assert solution.pacificAtlantic(heights) == [[0,4],[1,3],[1,4],[2,2],[3,0],[3,1],[3,2],[4,0],[5,0]]
> ```
>
> `Question 3: Computational Cost Breakdown`
>
> We appreciate your concerns about the computational costs between dLLM inference and AST operations. In our experiments, the inference time of the dLLM is roughly 10 times greater than the cost of the AST operations. That's to say, dLLM inference accounts for the majority of the time consumed.
>
> `Question 4: Performance Comparison`
>
> Please refer to our response to all reviewers.

---

### Author Response · Authors · 2025-11-23
**Response to All Reviewers**

Dear reviewers, thank you all for your thoughtful comments and for acknowledging our contributions. To address the common concerns and points of confusion raised by the reviewers, we would like to present the following explanations.

**Motivation and research question of our work**: As an emerging class of LLMs, dLLMs exhibit higher generation efficiency due to their unique advantage of parallel decoding. The primary motivation of our work is to leverage the advantage of dLLMs to **accelerate** UTG. Although dLLMs can decode multiple tokens in a single step, simply increasing the number of tokens decoded per step typically leads to a degradation in generation quality (e.g., decreased coverage). So the research question of our work is to address this trade-off with a novel method that increases the number decoded per step **without sacrificing output quality**.

**Comparison with AR models**: As described above, the motivation of this work is to explore how dLLMs, as a new type of model, can be leveraged to accelerate UTG. Our goal is not to generate better unit tests to achieve higher coverage. For this reason, the submission version did not include comparisons with existing auto-regressive (AR) models. However, we still provide results comparing our method with an AR model of comparable scale (Qwen-2.5-7B-Instruct).

Tables below demonstrate results compared with an AR LLM. In the table, each cell shows **average time** (s) / **average computational cost** (tflops), respectively. The results show the efficiency improvement compared with AR LLM in terms of generation speed and computational efficiency. We also provide a figure comparing Qwen-2.5-7B-Instruct with Dream-v0-7B + DIFFTESTER in the new version of our paper in Appendix C.2.1.

**python**

| n | Qwen-2.5-7B-Instruct | Dream-v0-7B-Instruct | Dream-v0-7B-Instruct + DIFFTESTER |
|:-:|:-------------------:|:-----------:|:-------------------------:|
| 1 | 4.41 / 530.88 | 4.99 / 338.53 | 3.33 / 192.71 |
| 2 | 8.25 / 1082.25 | 8.62 / 677.06 | 5.76 / 402.11 |
| 3 | 12.22 / 1673.15 | 12.18 / 1015.59 | 8.04 / 605.05 |
| 4 | 16.22 / 2261.25 | 15.78 / 1354.12 | 10.86 / 849.29 |
| 5 | 20.46 / 2881.37 | 19.39 / 1692.66 | 13.68 / 1093.38 |
| 6 | 24.43 / 3470.72 | 22.96 / 2031.19 | 16.94 / 1378.51 |
| 7 | 28.46 / 4068.15 | 26.53 / 2369.72 | 19.95 / 1644.05 |

**cpp**

| n | Qwen-2.5-7B-Instruct | Dream-v0-7B-Instruct | Dream-v0-7B-Instruct + DIFFTESTER |
|:-:|:-------------------:|:-----------:|:-------------------------:|
| 1 | 4.39 / 539.98 | 5.71 / 405.66 | 3.21 / 185.49 |
| 2 | 8.60 / 1149.27 | 10.11 / 811.32 | 5.56 / 384.08 |
| 3 | 12.93 / 1787.18 | 14.43 / 1216.98 | 7.96 / 588.17 |
| 4 | 17.83 / 2503.52 | 18.77 / 1622.64 | 10.65 / 816.77 |
| 5 | 22.08 / 3134.84 | 23.11 / 2028.30 | 13.33 / 1041.20 |
| 6 | 26.85 / 3838.59 | 27.38 / 2433.96 | 16.26 / 1294.46 |
| 7 | 32.46 / 4687.42 | 31.60 / 2839.62 | 19.97 / 1614.85 |

**java**

| n | Qwen-2.5-7B-Instruct | Dream-v0-7B-Instruct | Dream-v0-7B-Instruct + DIFFTESTER |
|:-:|:-------------------:|:-----------:|:-------------------------:|
| 1 | 7.03 / 868.45 | 5.87 / 419.79 | 3.53 / 211.23 |
| 2 | 13.48 / 1810.16 | 10.39 / 839.57 | 5.85 / 407.92 |
| 3 | 20.68 / 2869.84 | 14.92 / 1259.36 | 8.97 / 673.40 |
| 4 | 26.89 / 3792.26 | 19.96 / 1679.15 | 11.42 / 881.55 |
| 5 | 34.05 / 4851.20 | 24.22 / 2098.93 | 14.34 / 1130.74 |
| 6 | 41.68 / 5998.76 | 28.25 / 2518.72 | 17.55 / 1410.35 |
| 7 | 47.90 / 6927.35 | 32.64 / 2938.51 | 20.72 / 1678.98 |

**Why miss existing argumentation approaches on AR models**: In this experiment, the AR baseline employs a strictly vanilla decoding procedure—without any acceleration techniques (e.g., speculative decoding) or quality-enhancing strategies (e.g., CODAMOSA). The reason for not applying the **acceleration** techniques is that our work primarily aims to explore the fundamental advantages of dLLMs over AR models at the foundation model level. Applying additional acceleration strategies to AR models would compromise the fairness of comparison, and moreover, these techniques could also be applied to dLLMs in the future. The reason for not applying the **quality-enhancing** strategies is that our goal in this paper is to accelerate UTG rather than to produce higher-quality unit tests. Such quality-enhancing techniques can likewise be incorporated to improve the test cases generated by dLLMs.

Thank you for your time!

Best,

all the authors

---

### Author Response · Authors · 2025-11-25
**Response to All Reviewers**

Dear reviewers,

Thank you for recognizing the contributions of our work and for providing thoughtful comments and suggestions. We have carefully addressed the concerns and points of confusion raised in the reviews and provided detailed clarifications in our responses. We would like to kindly ask whether our explanations sufficiently resolve your concerns, or if there are any remaining points for which you would like further clarification or additional information. We are happy to provide more details if needed.

Thank you for your time!

Best,

all the authors

---

### Note · Authors · 2026-01-06

I have read and agree with the venue's withdrawal policy on behalf of myself and my co-authors.